# RADIAL BASIS OPERATOR NETWORKS

## ABSTRACT

Operator networks are designed to approximate nonlinear operators, which provide mappings between infinite-dimensional spaces such as function spaces. These networks are playing an increasingly important role in machine learning, with their most notable contributions in the field of scientific computing. Their significance stems from their ability to handle the type of data often encountered in scientific applications. For instance, in climate modeling or fluid dynamics, input data typically consists of discretized continuous fields (like temperature distributions or velocity fields). We introduce the radial basis operator network (RBON), which represents a significant advancement as the first operator network capable of learning an operator in both the time domain and frequency domain when adjusted to accept complex-valued inputs. Despite the small, single hidden-layer structure, the RBON boasts small $L^2$ relative test error for both in- and out-of-distribution data (OOD) of less than $1 \times 10^{-7}$ in some benchmark cases. Moreover, the RBON maintains small error on OOD data from entirely different function classes from the training data.

## 1 INTRODUCTION

### 1.1 BACKGROUND

Traditional feedforward neural networks (FNNs) and radial basis function (RBF) networks have been shown to be universal approximators of functions (Hornik et al., 1989; Park & Sandberg, 1991), meaning they are capable of representing the mapping between *finite* dimensional spaces. Thus, these networks are limited in their design to predicting a measurement acting on a subspace of $\mathbb{R}^d$ for some $d \in \mathbb{Z}^+$. Operator networks, however, are designed to learn the mapping between *infinite* dimensional spaces; they receive functions as input and produce the corresponding output function. Scientific computing has benefited from using operator networks to enhance or replace numerical computation for the purpose of simulation and forecasting on a wide array of applications to include computational fluid dynamics and weather forecasting (Azizzadenesheli et al., 2024).

The two primary neural operators that demonstrated immediate success are the deep operator network (DeepONet) (Lu et al., 2021) based on the universal approximation theorem in Chen and Chen (1995a), and the Fourier neural operator (FNO) (Li et al., 2021). The basic DeepONet approximates the operator by applying a weighted sum to the product of each of the transformed outputs from two FNN sub-networks. The upper sub-network, or *branch net*, is applied to the input functions while the lower *trunk net* is applied to the querying locations of the output function.

In contrast, the FNO is a particular type of Neural Operator network (Kovachki et al., 2023), which accepts only input functions (not querying locations for the output) and applies a global transformation on the function input via a more intricate architecture. Motivated by fundamental solutions to partial differential equations (PDEs), the FNO network sums the output of an integral kernel transformation to the input function with the output of a linear transformation. The sum is then passed through a non-linear activation function. To accelerate the integral kernel transformation, the FNO applies a Fourier transform (FT) to the input data, with the FT of the integral kernel assumed as trainable parameters.

Following their initial introduction, several extensions and modifications to FNO and DeepONet were introduced to improve performance in specific contexts. Examples include the Fourier-enhanced DeepONet (Zhu et al., 2023) to improve DeepONet's robustness against Gaussian noise,

U-FNO (Wen et al., 2022) and MIONet (Jiang et al., 2024) introduce U-Net paths into the Fourier layer of the FNO architecture to improve accuracy for multi-phase flow applications, and model-parallel FNOs (Grady et al., 2023) parallelise the structure of FNO to reduce computation load for high-dimensional data. However, many of these situational improvements did not result in clear error reductions across a variety of contexts, at least not enough to justify the additional complexity in architecture contained in some of the proposed methods. This has changed with the recent introduction of a new neural operator.

The Laplace Neural Operator (LNO) (Cao et al., 2024) has recently become a benchmark standard for operator networks due to its improved handling of transient responses and non-periodic signals, limitations inherent in the Fourier Neural Operator (FNO). LNO achieves this by leveraging the pole-residue method to represent both transient and steady-state responses in the Laplace domain, leading to better test performance on out-of-distribution (OOD) data in most contexts. Additionally, LNO boasts a reduced training cost and a simpler network architecture. For these reasons, we have selected LNO as the primary comparison for our new operator network, alongside FNO and DeepONet. To thoroughly evaluate performance, we include a problem scenario from (Cao et al., 2024) that highlights LNO's small OOD error in predictions.

### 1.2 OUR CONTRIBUTIONS

We propose the radial basis operator network (RBON) based on the universal approximation theorem in Chen and Chen (1995b); a novel operator network that is, to the best of our knowledge, the first to be entirely represented with radial basis functions.

- The universal approximation result in Chen and Chen (1995b) is extended to normalised RBONSs (NRBONs).
- The RBON is the first network to successfully learn an operator entirely in both the time domain and frequency domain, by altering the algorithm to accept complex data types.
- Despite the simple single-hidden-layer structure, the particular implementation of the RBON within demonstrates impressively small error on both in-distribution (ID) and OOD data, outperforming LNO by several orders of magnitude.
- The RBON demonstrates successful results on the first OOD example where the OOD input is an entirely different base function. Typically, OOD input functions for introducing new operator networks are a scaling, shifting, or simple transformation of the input functions used in training.

While operator networks are usually tested only using data generated from known systems, such as in systems of partial differential equations (PDEs), we include a scientific application where the data is real physical measurements and the underlying operator is unknown. This demonstrates the ability of RBON to make accurate forecasts for time-dependent systems, for the purposes of scientific experimentation. The rest of the paper is organised as follows, the theoretical foundation and details regarding the particular implementation are presented in Section 2, which precedes the results of the numerical experimentation first on generated data followed by the observed data in 3, with the discussion and conclusion at the end.

## 2 METHODOLOGY

The RBON is a numerical representation, $G_\dagger$, for an operator, $G : \mathcal{U} \to \mathcal{V}$, where $\mathcal{U}$ and $\mathcal{V}$ are infinite dimensional spaces, using radial basis functions. Following the work as shown in Chen and Chen (1995b), we present, without proof, the universal approximation theorem for such a representation as well as extending the theorem to include NRBONs. The subsequent section details the precise implementation used for the experimental results.

### 2.1 THEORETICAL FOUNDATION

In distribution theory the Schwartz space, $\mathcal{S}(\mathbb{R}^d)$, is the space of rapidly decaying functions that are infinitely differentiable and whose derivatives decay faster than a polynomial. Essentially, these

are smooth functions that vanish quickly away from their center. The space containing all linear functionals that act on the Schwartz space is referred to as the space of tempered distributions and is represented symbolically as $\mathcal{S}'(\mathbb{R}^d)$; the prime notation connotes the duality relationship between the spaces. These spaces are for defining the necessary regularity for the radial basis functions used in the approximation.

Noting that $C(A)$ represents all continuous functions defined on $A$, consider the functions $g$ such that,

$$g \in C(\mathbb{R}) \cap S'(\mathbb{R}). \tag{1}$$

meaning $g$ is in the space of tempered distributions and is continuous on $\mathbb{R}$. Choosing $||\boldsymbol{x}||_{\mathbb{R}^d}$ to represent the Euclidean norm for $x \in R^d$, we can represent a radial basis function acting on $\boldsymbol{x}$ as

$$g(\lambda||\boldsymbol{x} - \mu||_{\mathbb{R}^d})$$

for constants $\lambda \in \mathbb{R}, \mu \in \mathbb{R}^d$. Then we have the following (see Chen & Chen (1995b) for the proof with details).

**Theorem 2.1** *Suppose $g$ is not an even polynomial and satisfies equation 1, $X$ is a Banach space where $K_1 \subseteq X, K_2 \subseteq \mathbb{R}^d$ are two compact sets in $X$ and $\mathbb{R}^d$ respectively. Suppose also that $\mathcal{U}$ is a compact set in $C(K_1)$, $G$ is a nonlinear continuous operator, mapping $\mathcal{U}$ into $C(K_2)$, then for any small positive $\epsilon$, there are positive integers $M, N, m$, constants $\xi_i^k, \omega_k, \lambda_i \in \mathbb{R}, k \in \{1, \dots, N\}, i \in \{1, \dots, M\}$, $m$ points $\boldsymbol{x}_1, \dots, \boldsymbol{x}_m \in K_1, \boldsymbol{c}_1, \dots, \boldsymbol{c}_N \in \mathbb{R}^d$, such that*

$$\left| G(u)(\boldsymbol{y}) - G^\dagger(u^m)(\boldsymbol{y}) \right| < \epsilon$$

*for every $u \in \mathcal{U}$ and $\boldsymbol{y} \in K_2$, where $u^m = (u(\boldsymbol{x}_1), \dots, u(\boldsymbol{x}_m))$, and*

$$G^\dagger(u^m)(\boldsymbol{y}) = \sum_{i=1}^{M} \sum_{k=1}^{N} \xi_i^k g(\lambda_i||u^m - \mu_{ik}^m||_{\mathbb{R}^m}) g(\omega_k||\boldsymbol{y} - \boldsymbol{c}_k||_{\mathbb{R}^d}) \tag{2}$$

*for $\mu_{ik}^m = (\mu_{1k}^m, \dots, \mu_{mk}^m), k = 1, \dots, N$.*

For $\epsilon$ and $\xi_i^k$ given as in Theorem 2.1 set

$$\tilde{\xi}_i^k = \xi_i^k \sum_{i=1}^{M} \sum_{k=1}^{N} g(\lambda_i||u^m - \mu_{ik}^m||_{\mathbb{R}^m}) g(\omega_k||\boldsymbol{y} - \boldsymbol{c}_k||_{\mathbb{R}^d}), \tag{3}$$

and the corollary extending the theorem for the normalised representation follows immediately.

**Corollary 2.1.1** *Under the same assumptions in Theorem 2.1, and with $\xi_i^k$ as defined in equation 3, we have*

$$\left| G(u)(\boldsymbol{y}) - \tilde{G}^\dagger(u^m)(\boldsymbol{y}) \right| < \epsilon$$

*where*

$$\tilde{G}^\dagger(u^m)(\boldsymbol{y}) = \sum_{i=1}^{M} \sum_{k=1}^{N} \tilde{\xi}_i^k \frac{g(\lambda_i||u^m - \mu_{ik}^m||_{\mathbb{R}^m}) g(\omega_k||\boldsymbol{y} - \boldsymbol{c}_k||_{\mathbb{R}^d})}{\sum_{i=1}^{M} \sum_{k=1}^{N} g(\lambda_i||u^m - \mu_{ik}^m||_{\mathbb{R}^m}) g(\omega_k||\boldsymbol{y} - \boldsymbol{c}_k||_{\mathbb{R}^d})}. \tag{4}$$

The RBON, as represented in equation 2, comprises two single-layer sub-networks of radial basis functions. This architecture extends the concept of RBF networks to operators, analogous to how DeepONet extended FNNs. The sub-network that processes the function input $u^m$ is called the branch net. Here, $u^m$ represents the input function $u$ sampled at $m$ point locations, as defined in the theorem. The trunk net, on the other hand, receives inputs corresponding to the domain locations where the network will produce output function values. The diagram in Figure 1 shows the basic RBON architecture with the additional linear output transformation, $\mathcal{L}$.

## 2.2 PRACTICAL IMPLEMENTATION

Having established the theoretical foundation, we now turn to the practical implementation of our approach. This section outlines the step-by-step process for the realised implementation of both

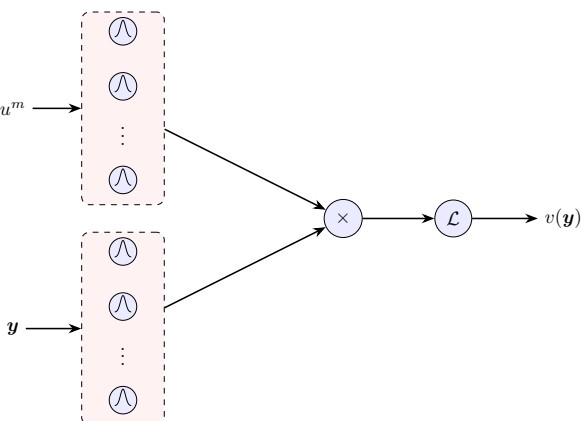

Figure 1: The upper network is the branch net that receives the function input, and the lower trunk net receives the query location input. The product between the trunk and branch networks is represented by the Kronecker product, and $\mathcal{L}$ denotes a linear transformation.

RBON and NRBON. The implementation consists of several key steps that translate our theoretical model into a functional algorithm.

From Theorem 2.1, recall $u^m \in \mathbb{R}^m$ represents the numerical approximation of the function $u$ sampled at $m$ locations, $G^\dagger$ is the network approximation of the operator, $G$, mapping $u^m$ to the function $v$ at the query location $\boldsymbol{y} \in R^d$. Then, given input functions $u_j^m$ for $j \in \{1, \ldots, J\}$, and query locations $\boldsymbol{y}_\ell$ for $\ell \in \{1, \ldots, L\}$ where $J$ and $L$ denote the number of training input functions and query points, respectively, we outline the process for finding the network parameters.

**RBF transformations.** In both the trunk and branch networks we employ Gaussian functions for the RBF transformations, defined as

$$\phi(\mathbf{x}, \mathbf{c}, \sigma) = \exp\left(-\frac{||\mathbf{x} - \mathbf{c}||^2}{2\sigma^2}\right)$$

where $\mathbf{c}$ and $\sigma$ are the RBF centers and spreads. The RBF centers are determined using K-means clustering (Lloyd, 1982; Forgy, 1965) on the input data for each sub-network, with the spreads calculated based on inter-cluster distances. The branch and trunk network transformations on an input pair $\{u_j^m, \boldsymbol{y}_\ell\}$, with $M$ and $N$ RBFs, are represented by the vectors

$$\mathbf{b}(u_j^m) = [\phi(u_j^m, \mathbf{c}_1^b, \sigma_1^b), ..., \phi(u_j^m, \mathbf{c}_M^b, \sigma_M^b)]^T$$
$$\mathbf{t}(\boldsymbol{y}_\ell) = [\phi(\boldsymbol{y}_\ell, \mathbf{c}_1^t, \sigma_1^t), ..., \phi(\boldsymbol{y}_\ell, \mathbf{c}_N^t, \sigma_N^t)]^T$$

where $\mathbf{c}_i^b, \mathbf{c}_k^t$ are the RBF centers and $\sigma_i^b, \sigma_k^t$ are spreads for the associated branch and trunk networks.

**Weight parameter calculation.** For each query location $\boldsymbol{y}_\ell$, we first compute

$$\boldsymbol{\Phi}_\ell = [\mathbf{b}(u_1^m) \otimes \mathbf{t}(\boldsymbol{y}_\ell), ..., \mathbf{b}(u_J^m) \otimes \mathbf{t}(\boldsymbol{y}_\ell)]$$

where $\otimes$ denotes the Kronecker product, making $\boldsymbol{\Phi}_\ell$ of dimension $NM \times J$. The weights $\boldsymbol{\xi}_\ell$ of dimension $NM \times 1$, are then determined by solving

$$\boldsymbol{\xi}_\ell^T \boldsymbol{\Phi}_\ell = [v_1(\boldsymbol{y}_\ell), \cdots, v_J(\boldsymbol{y}_\ell)]$$

using the Moore-Penrose inverse (Moore, 1920; Penrose, 1955). This process yields $L$ weight vectors $\boldsymbol{\xi}_\ell$, for each query point. The final weight vector $\boldsymbol{\xi}$ is obtained by element-wise averaging across the $L$ vectors $\boldsymbol{\xi}_\ell$. Given the input $u^m$, the network approximation for the associated output function $v$ at query point $\boldsymbol{y}$ is then

$$G^\dagger(u^m)(\boldsymbol{y}) = \mathcal{L}(\boldsymbol{\xi}^T[\mathbf{b}(u^m) \otimes \mathbf{t}(\boldsymbol{y}_\ell)])$$

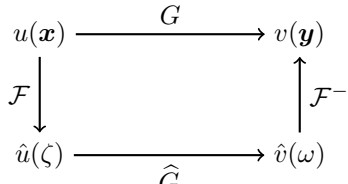

Figure 2: The RBON is capable of learning either $G$ or $\widehat{G}$ where $\mathcal{F}$ denotes the Fourier transform and $\zeta, \omega$ are the frequency input.

where $\mathcal{L}$ denotes a linear transformation applied to the final output whose parameters are solved for directly using the training data.

**NRBON modification.** The NRBON differs from RBON in normalizing the products of the branch and trunk outputs by dividing each element of the vector $[\mathbf{b}(u^m) \otimes \mathbf{t}(\boldsymbol{y}_\ell)]$ by the vector's sum. This normalization adjusts the computation of $\boldsymbol{\Phi}_\ell$ by its column totals.

Using K-means to determine the parameter locations for the RBFs limits the number of RBFs in the representation by the size of the training data set. It is worth noting that manually assigning the centers for the RBFs produces satisfactory results, but tends to result in larger error than using K-means. Hence manually assigning centers is only advisable when working with small training sets. Moreover, the majority of the variation in train/test error is mostly due to the varying results from the location parameters determined by the K-means clustering.

Concluding the description of the practical implementation, we note that the network weights can be solved for using an iterative approach such as least-mean-squares, but results in weights that on average produce larger error in their predictions.

### 2.3 LEARNING IN THE FREQUENCY DOMAIN

The RBON is designed to learn the operator in the frequency domain as well as in the time domain. The frequency domain is a representation of signals or functions in terms of their frequency components, rather than time. It allows analysis of how signals vary with frequency, providing insight into characteristics like energy, power, and periodicity. The frequency domain is often used to examine cyclic behavior, separate overlapping signals, and simplify certain mathematical operations on signals. Considering that functions have a global representation in the frequency domain, this can have benefits in reducing the variability on the RBONs predictions for OOD data.

Thus, the RBON can be trained on functions in the time domain to approximate the operator $G$ shown in Figure 2, or the Fourier transform, $\mathcal{F}$, can be used to convert functions to the frequency domain, in which case the RBON is learning the approximation for what is labeled as $\widehat{G}$ in figure 2. This is especially beneficial for applications where the data is stored in the frequency domain representation.

While algorithmically the RBON follows the same process whether training in the time domain or frequency domain, we denote the application of the RBON to the frequency domain as F-RBON. Numerically, the coded functions for F-RBON accept complex-valued input since functions defined on the frequency domain are stored numerically using complex valued arrays. This allows the F-RBON to capture both magnitude and phase information inherent in frequency domain representations.

## 3 NUMERICAL EXPERIMENTATION AND RESULTS

This section is partitioned into *numerical computing* experiments and a scientific application based solely on data collected from observed measurements. This demonstrates the ability for the RBON to learn the mapping in a variety of contexts including when the mathematical representation for the operator is unknown. We define the numerical computing setting here as scenarios where the data is completely generated from numerical approximations of solutions to mathematical equations. Thus,

the operator is known precisely and the results of the RBON can be compared to the numerical approximation of the operator output.

Alternatively, the governing equations for scientific applications are not always known and data is often aggregated from physical measurements. Distinguishing between settings using *generated* data as opposed to *observed* data shows the flexibility of the RBON and its ability to support scientific experimentation and forecasting.

For all the numerical computing experiments, we limited the size of the trunk and branch networks to be no larger than 15 nodes each, capping the number of multiplier parameters in the hidden layer at 225. These restrictions demonstrate the network's ability to maintain small errors even under incredibly strict size constraints. All code for the RBON learning representation was implemented using the Julia programming language (Bezanson et al., 2017), chosen for its high-performance numerical computing capabilities, and has been made available at `https://anonymous.4open.science/r/RBON-C1D0/`

### 3.1 NUMERICAL COMPUTING

In each of the following settings, there is a governing system of PDEs defined on a spatio-temporal domain, $\Omega \equiv (0, T) \times (0, L)$ for some final time $T > 0$ and length $L > 0$. The operator network, $G^\dagger$, was trained to learn an operator $G$ within the PDE framework that maps functions representing the initial state or forcing term to the solution over the entire domain.

The input functions for the network for the in distribution data will thus be a family of functions representing an initial state (or forcing term) and parameterized across a specified range of values. ID data was segmented to produce a validation and test set. The validation set was used to optimized the size of the network over a few selected options. The test set provides the in-distribution test error with the out-of-distribution errors based on a set of input functions that have been more significantly altered from the in distribution data.

#### 3.1.1 WAVE EQUATION

Consider the wave equation of the form

$$\frac{\partial^2 u}{\partial t^2} = c^2 \frac{\partial^2 u}{\partial x^2} \quad \text{for } (t, x) \in \Omega,$$

where $c$ is the speed of propagation of the wave and $u(t, x)$ models the displacement of a string with Dirichlet boundary conditions. The operator network, $G^\dagger$, was trained to learn the mapping, $G$, from the initial state to the solution, $G\colon u_0(x) \to u(x, t)$. For the ID data, we particularize the initial condition as

$$u_0(x) = 2e^{-\left(x - \frac{L}{2}\right)^2} + \frac{ax}{L}$$

where $a$ is parameterized across the range $[1, 4]$ with step size 0.001. The OOD test set uses the same base function for $u_0(x)$, but for values of $a$ in the range $[5, 5.5]$.

#### 3.1.2 BURGERS EQUATION

Consider the well-known Burgers' equation

$$\frac{\partial u}{\partial t} + u \frac{\partial u}{\partial x} = \nu \frac{\partial^2 u}{\partial x^2}, \quad \text{for } (t, x) \in \Omega,$$

subject to homogeneous Dirichlet boundary conditions and under the following initial conditions $u_0(x) = a \sin \pi x$ where $a$ ranges across the interval $[0.1, 5]$. The operator learned is thus $G\colon u_0(x) \to u(x, t)$ and the RBON is tested on the set of polynomial functions $u_0(x) = bx(x - 1)$ where $b$ is in the range $[3.5, 4.5]$ for the OOD data. Successful testing on the polynomial input after only being trained on the sine function is quite remarkable. The numerical data was generated using the exact solutions as derived in Öziş et al. (2003).

Table 1: Average relative $L^2$ error on test data reported with margin of error in parentheses.

| Network | In/Out | Wave | Burgers | Beam |
|---|---|---|---|---|
| **RBON** | In | 9.4E−4(4.9E−5) | 3.6E−3(6.0E−4) | **4.1E−8(3.3E−6)** |
| | Out | 1.0E−1(2.0E−3) | 2.6E−1(1.3E−2) | **1.5E−8(2.5E−7)** |
| **NRBON** | In | 1.2E−5(9.4E−7) | **3.3E−3(9.0E−4)** | 1.6E−7(2.4E−7) |
| | Out | 3.2E−1(1.1E−2) | 1.0E−1(5.7E−3) | 2.0E−8(4.9E−9) |
| **F-RBON** | In | **3.0E−6(2.2E−7)** | 5.9E−3(1.1E−3) | 1.1E−1(1.3E−1) |
| | Out | **8.6E−3(1.7E−4)** | 2.3E−2(5.5E−3) | 6.6E−2(7.0E−3) |
| **LNO** | In | 5.6E−1(1.1E−3) | 1.7E−1(4.3E−4) | 1.0E−2(3.9E−3) |
| | Out | 5.9E−1(9.2E−4) | 2.0E−1(8.0E−6) | 6.8E−3(1.5E−3) |
| **FNO** | In | 9.9E−4(2.3E−5) | 9.3E−3(1.2E−3) | 4.0E−3(6.1E−3) |
| | Out | 1.1E−1(1.4E−3) | **1.7E−2(7.0E−6)** | 1.5E−3(2.2E−4) |
| **DeepONet** | In | 5.3E−2(2.5E−4) | 9.9E−1(4.0E−5) | 2.9E−1(2.9E−1) |
| | Out | 4.9E−2(3.4E−5) | 9.9E−1(2.0E−6) | 2.5E−1(1.4E−2) |

### 3.1.3 EULER-BERNOULLI BEAM EQUATION

The Euler-Lagrange equation for a dynamic Euler-Bernoulli homogeneous beam is

$$EI\frac{\partial^4 u}{\partial x^4} + \rho A \frac{\partial^2 u}{\partial t^2} = f(t,x) \text{ for } (t,x) \in \Omega,$$

where $E$ is Young's modulus, $I$ is the second moment of the area of the beam's cross section. The beam's density is denoted by $\rho$ and the cross-sectional area as $A$. The operator network learns the mapping from the source term to the solution: $G\colon f(t,x) \to u(t,x)$. For the ID data we particularize the source term as $f(t,x) = ae^{-0.05x}(1-10^2)\sin(10t)$ for $a$ in $[0.05, 10]$. The source term for the out of distribution data is $f(t,x) = ae^{-x}(1-10^2)\sin(10t)$ for $a$ in $[1.24, 10.19]$. This scenario was use for testing in (Cao et al., 2024). We include it here for benchmarking purposes, but note we decrease the size of the training set to include in-distribution test error.

### 3.1.4 RESULTS

The results from these experiments can be seen in Table 1, which displays the average $L^2$ error for each function in the ID and OOD test sets. The $L^2$ relative error is computed as follows

$$L^2 \ rel. \ error = \frac{||v^{true} - v^{pred}||_2}{||v^{true}||_2},$$

where $v^{\text{true}}$ represents the ground truth values at all query points in the space-time domain, and $v^{\text{pred}}$ represents the network's predictions at the same locations.

The margin of error (MOE), shown in parentheses, was computed by multiplying the standard error of the point estimate by the critical value for a 95

Across the majority of problems, the RBON variants outperform the LNO, with the NRBON achieving consistently superior performance on both ID and OOD data. Overall, RBON variants collectively tend to outperform other operator networks, with one exception. Notably, operator networks generally exhibit smaller errors for the Beam equation due to their ability to accurately represent linear operators. Networks that leverage global representations—such as FNO, LNO, and F-RBON (which trains on data with global representations)—tend to generalize better, while other networks overfit ID data and have significantly worse performance on OOD data. This difference is especially noticeable with the OOD input data for the Wave problem due to its highly oscillatory behavior.

DeepONet initially suffered from overfitting, resulting in poor OOD performance, but early stopping significantly improved its OOD errors, albeit at the cost of slightly worse ID errors. However, this improvement came at the expense of efficiency: DeepONet required significantly larger sub-networks, with over 10,000 products between trunk and branch outputs, compared to fewer than 200 products in the RBONs.

## 3.2 SCIENTIFIC APPLICATION

Modeling the relationship between atmospheric $CO_2$ and global temperature is a complex process involving a large number of variables with many of them potentially unknown (Mills et al., 2019). Focusing specifically on an operator that does not have a well-defined mathematical representation, we demonstrate the capacity of the RBON to learn the mapping between monthly atmospheric $CO_2$ measurements and both local and average global monthly temperatures. This provides a template for prediction and forecasting with the RBON based on collected data.

For this section, the RBON is used to learn the operators

$$G_{avg} \colon u(t) \to T^{avg}(t),$$
$$G_{loc} \colon u(t) \to T^{loc}(t)$$

where $u$ represents the atmospheric $CO_2$ defined for $t$ in a given time interval, and $T^{avg}$ represents the average global temperature as published in Our World in Data (2023). The function $T^{loc}$ is local temperature readings at the same site location where the $CO_2$ data was collected. Specifically, atmospheric $CO_2$ concentrations (ppm) derived from in situ air measurements at the well known Mauna Loa, Observatory, Hawaii (Keeling et al., 2001). The local temperature readings are much more variable than the global average and hence less easily predicted.

The nature of the operators $G_{avg}$ and $G_{loc}$ is expected to evolve in time due to fluctuations in other contributing factors, however, when continuously updating the RBON with new data, the predictions become quite accurate. While, it is possible to feed the $CO_2$ readings into the network as one function, the centers for the RBFs must be set manually as the K-means algorithm requires at least two function inputs. Instead, it is preferable to parameterize the functions across the years such that $t \in \{1, 2, \dots, 12\}$ with each number corresponding to the month of the year the measurement was taken. Then operators are thus more accurately represented as

$$G_{avg} \colon u_n(t) \to T_n^{avg}(t),$$
$$G_{loc} \colon u_n(t) \to T_n^{loc}(t)$$

where $n$ corresponds to a specific year. Training on the historical data, omitting years with incomplete data, yields remarkable accuracy in the RBON predictions as shown in Table 2.

### 3.2.1 RESULTS

The results in Table 2 highlight the effectiveness of RBON in accurately predicting both local monthly average temperatures and global average temperatures. To evaluate the forecasting accuracy, we trained RBON and NRBON networks on historical temperature data, withholding the most recent two or five years from the training set for testing. In addition to these models, we compared their performance against LNO, DeepONet, FNO, and LSTM. This comprehensive evaluation demonstrates the robustness of RBON across diverse benchmarks, including traditional time-series approaches such as LSTM Hochreiter & Schmidhuber (1997) and as well as other operator networks.

Based solely on monthly $CO_2$ measurements and the month encoding for querying the output temperature, the RBON maintains an $L^2$ relative error of less than $10\%$, with NRBON performing similarly. Figure 3 displays a comparison between the trained RBON networks' global temperature predictions and actual global temperature readings. The left graph shows results when holding out the most recent two years, while the right graph illustrates the outcome when holding out the most recent five years of data. Interestingly, several networks—including RBON, F-RBON, DeepONet and LSTM—performed similarly on the smoother global temperature data. However, performance on the more variable local temperatures at the observatory publishing the atmospheric $CO_2$ measurements (Keeling et al., 2001) provided a more clear distinction as RBON outperformed other networks, which struggled to capture the finer-scale variations in the data. Figure 4 provides the

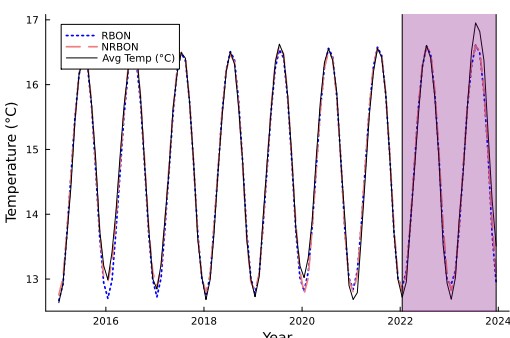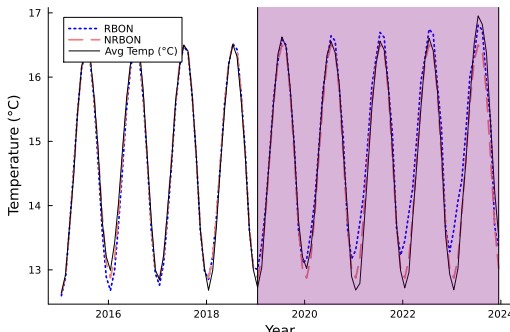

Figure 3: Two (left) and five (right) year average global temperature predictions based on $CO_2$ input. Forecast values are in the shaded region.

Table 2: Average relative $L^2$ test error on local and global temperature data as predicted based on atmospheric $CO_2$ input data.

|  | RBON | NRBON | F-RBON | LNO | LSTM | FNO | DeepONet |
|---|---|---|---|---|---|---|---|
| **Global temp: 2 yr** | 0.02 | 0.14 | 0.02 | 0.96 | 0.02 | 0.31 | **0.01** |
| **5 yr** | 0.02 | 0.15 | **0.01** | 0.97 | 0.02 | 0.44 | **0.01** |
| **Local temp: 2 yr** | **0.07** | 0.13 | 0.04 | 0.94 | 0.35 | 0.18 | 0.15 |
| **5 yr** | **0.07** | 0.13 | 0.13 | 0.95 | 0.51 | 0.22 | 0.14 |

visual comparison for local temperatures versus the predictions from the RBON variants. Note that temperature data for the local set was only available through 2018.

The significance of this result implies a robust model capable of providing reliable future temperature projections based on various atmospheric $CO_2$ scenarios under different climate responses. This robustness stems from the model's ability to isolate the impact of $CO_2$ on temperature, as the effects of other contributing elements are learned in the operator approximation. While predicting solely based on $CO_2$ measurements provides a simple example, there is an opportunity to include other contributing factors in the operator input to understand how co-variation among several input variables may affect the output.

Testing revealed that increasing the width of the branch and trunk networks enhances the model's flexibility to match highly variable and erratic behavior. However, given highly oscillatory data, the plain RBON can occasionally produce peaks and valleys that deviate too far from the data range when increasing model width. In contrast, the NRBON can increase its network size without generating extreme peaks. Consequently, the smaller RBONs used yield a more stable regression appearance, while the larger NRBON networks produce outputs that attempt to capture more of the random extreme values. This results in a slightly higher error ($\leq 0.17$) for the NRBON, but a shape that more closely resembles the true graph. Figures displaying the models' output over a larger time interval, providing a clearer picture of this phenomenon upon close inspection, can be viewed in the Appendix. This difference is particularly evident in the right graph of Figure 6, where the NRBON attempts to better match the variability in the peaks, while the RBON produces a more steady-state prediction when trained on the smaller set of historical data.

For completeness, we include all results pertaining to learning the operator in the frequency domain, namely the F-RBON. These results are presented in Table 2, and the corresponding plots can be viewed in Section A.1 of the Appendix. It's worth noting that this dataset does not naturally lend itself to a Fourier transform, and the additional computational work is unnecessary since the representation in the time domain is sufficient.

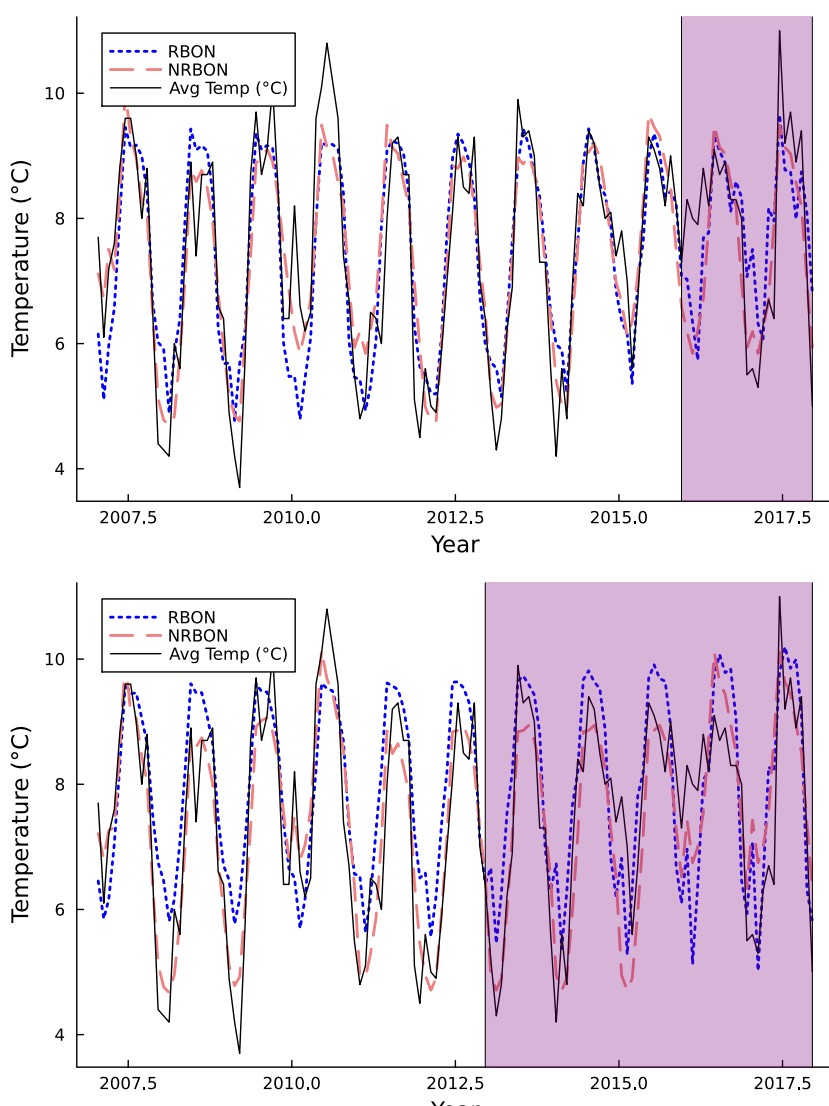

Figure 4: Two (top) and five (bottom) year local temperature predictions based on $CO_2$ input. Forecast values are in the shaded region.

## 4 DISCUSSION AND CONCLUSION

The RBON and its variants offer a simple yet powerful network architecture with prediction capabilities that yield errors smaller than the current leading operator network. The network's compact size provides opportunities for enhanced interpretability and reduced computational load, allowing for exact solutions of network parameters. Most variation across training cycles arises from the location and scale parameters of the RBFs, largely due to K-means' tendency to converge on local extrema. This variability can lead to errors differing by several orders of magnitude between runs of the K-means algorithm. A practical solution is to run K-means multiple times and select the configuration that minimizes the overall within-cluster distances. Furthermore, the RBON serves as an excellent tool for scientific computing, where recent advancements have only begun to explore the potential of operator networks in various fields. Finally, the RBON's ability to train on both real and complex-valued inputs, combined with its other strengths, makes it a promising candidate for applications in signal processing and computer vision tasks.

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

# A APPENDIX

## A.1 TEMPERATURE MODEL VISUALS

### A.1.1 RBON AND NRBON VISUALS

This section provides visuals for the temperature data and the NRBON and RBON predictions based on input $CO_2$ data on a larger time axis to provide a more complete view of the overarching trends. Specifically, Figures 5 and 6 provide the global and local temperature predictions over a much larger time period than the graphs present in Section 3.2.1.

### A.1.2 F-RBON VISUALS

This section of the Appendix provides the visual results based on the F-RBON model input for the temperature data discussed in Section 3.2.1 of the paper. Figure 7 displays the average monthly global temperature versus the model prediction for an F-RBON based on $CO_2$ measurement input after training on historical data, omitting the most recent two (left graph) and five (right graph)

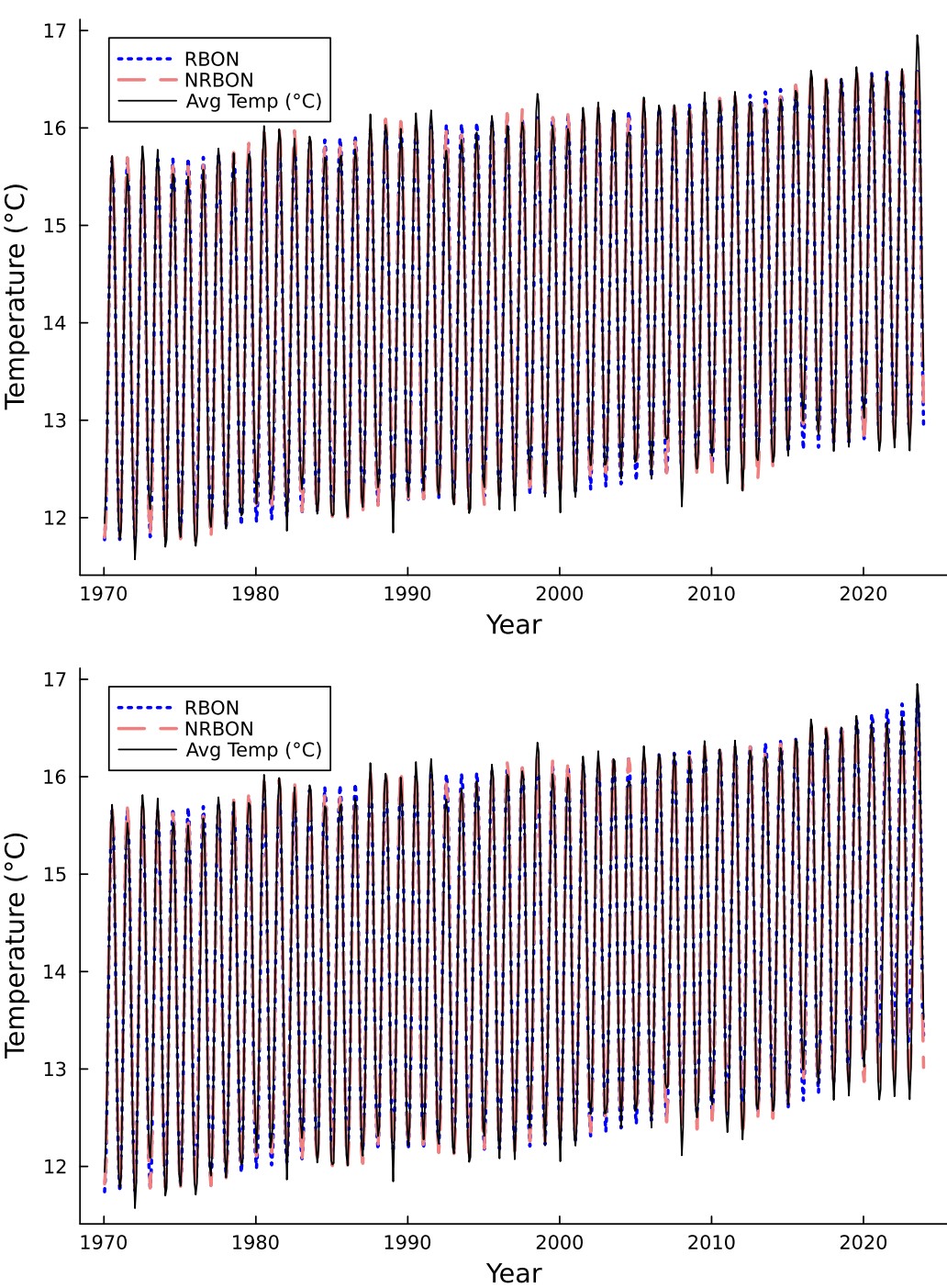

Figure 5: Two (top) and five (bottom) year average global temperature predictions based on $CO_2$ input.

temperature data. Figure 8 displays the same, but displays the results over a longer time interval to capture the complete overall trend.

The figures 9 and 10 show the F-RBON comparison on the local temperature data on both the shorter time axis in 9 versus the longer axis in 10.

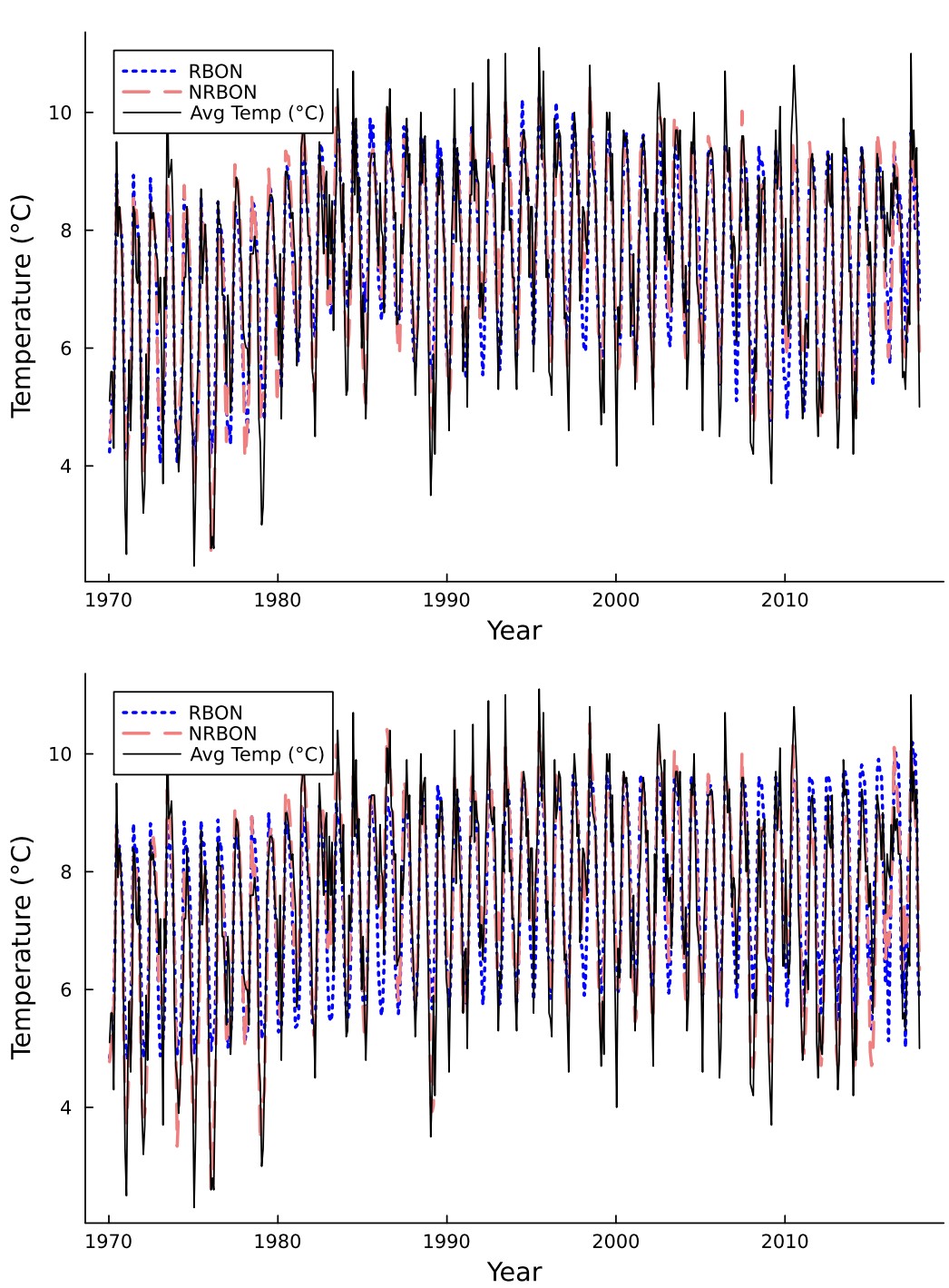

Figure 6: Two (top) and five (bottom) year average local temperature predictions based on $CO_2$ input.

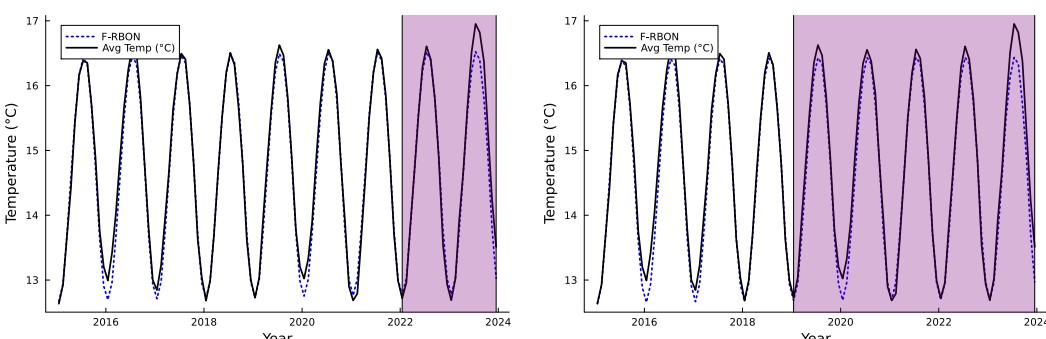

Figure 7: Two (left) and five (right) year average global temperature predictions based on $CO_2$ input.

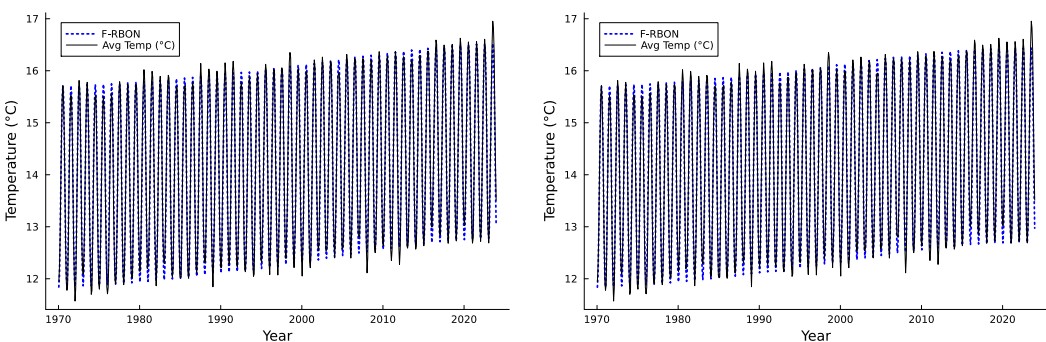

Figure 8: Two (left) and five (right) year average local temperature predictions based on $CO_2$.

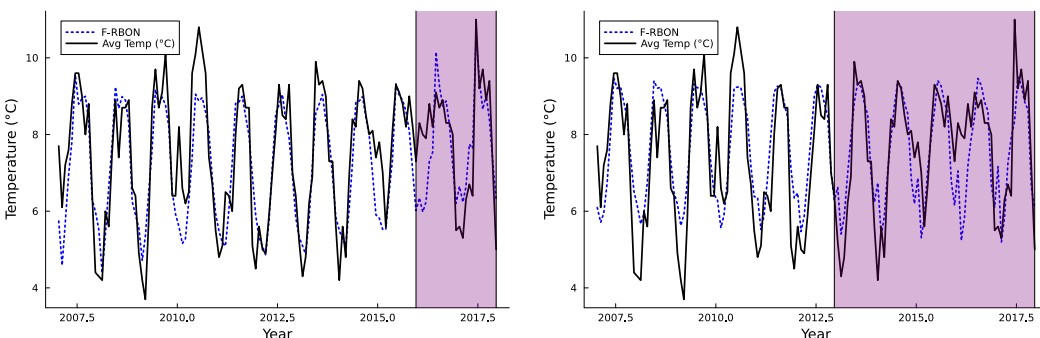

Figure 9: Two (left) and five (right) year average global temperature predictions based on $CO_2$ input.

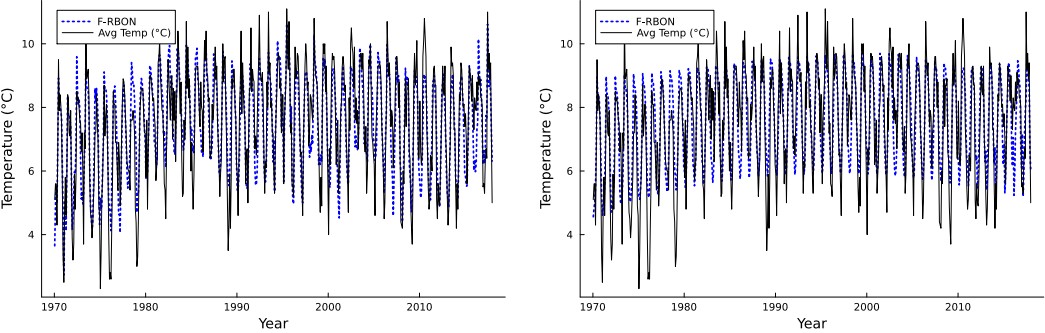

Figure 10: Two (left) and five (right) year average local temperature predictions based on $CO_2$.

