# OpenReview forum: "Radial Basis Operator Networks"
_ICLR.cc/2025/Conference — ICLR 2025 Conference Withdrawn Submission_

### Official Review · Reviewer_YY4J · 2024-10-20

**Soundness:** 3
**Presentation:** 2
**Contribution:** 2
**Rating:** 5
**Confidence:** 3

**Summary:**

This paper introduces a Radial basis operator network for mapping infinite dimensional spaces like function spaces. This approach can model temporal as well as frequency information. The introduction gives a good motivation for the problem, the literature is vast, and the methods are explained well. The paper shows the efficacy of the proposed approach on a weather dataset.

**Strengths:**

The paper is written.
Theory enhances the paper.

**Weaknesses:**

The experiments and comparisons are a bit small and need to be extended to understand the efficacy of RBON.

**Questions:**

•	Weather data is simple. Can more complex data be added to the comparison as well? There are many time series benchmark data sets (https://arxiv.org/pdf/2303.06053).
•       More comparison needs to be added. Like LNO, FNO, LSTM, etc. This will give a holistic view wrt performance.
•	Any insights on why the model did not perform as well as global in the local prediction? Is it because of short-term fluctuations or long-term trends?
•	Figures 5 and 6 are too compact and don’t add much information. Can be added to the appendix. A zoomed-in version of the plot would be helpful.
•	A sensitivity study for k means for deciding centers for RBFs would be useful. Showing the effect of varying values of k on model performance will give many insights.

---

> ### Author Response · Authors · 2024-11-26
>
> Dear Reviewer YY4J,
>
> Thank you for your insights, we believe the revised manuscript is much improved based on your suggestions. Below, we address your comments point by point:
>
> **• Weather data is simple. Can more complex data be added to the comparison as well? There are many time series benchmark data sets ([arXiv link](https://arxiv.org/pdf/2303.06053)).**
>
> We appreciate your suggestion to include more complex datasets. While we focused our efforts on extensive numerical experiments for the PDE problems, which are inherently more complex than the time-series example, we agree that exploring additional datasets would further strengthen our study. Due to time constraints, we could not include this in the current manuscript. However, we plan to incorporate these benchmark datasets, including those suggested, in future work to validate RBON’s generalizability in functional time forecasting.
>
> ---
>
> **• More comparison needs to be added. Like LNO, FNO, LSTM, etc. This will give a holistic view wrt performance.**
>
> Thank you for pointing this out. In response, we have extended our comparisons to include LNO, FNO, DeepONet, and LSTM. These additional experiments provide a broader perspective on the performance of RBONs relative to benchmark operator networks and time-series models. The results have been summarized in the revised manuscript.
>
> ---
>
> **• Any insights on why the model did not perform as well as global in the local prediction? Is it because of short-term fluctuations or long-term trends?**
>
> This is an important observation. The reduced performance in local predictions is primarily due to short-term fluctuations, which dominate local datasets (more random variation). Global predictions, on the other hand, benefit from averaging effects over larger scales, which smooth out these fluctuations (just seasonality and trend). We have replaced the local temperature figures with zoomed-in versions to better highlight this.
>
> ---
>
> **• Figures 5 and 6 are too compact and don’t add much information. Can be added to the appendix. A zoomed-in version of the plot would be helpful.**
>
> Thank you for this feedback. We have moved Figures 5 and 6 to the appendix and replaced them with zoomed-in versions.
>
> ---
>
> **• A sensitivity study for k-means for deciding centers for RBFs would be useful. Showing the effect of varying values of k on model performance will give many insights.**
>
> We agree that a sensitivity analysis would provide valuable insights. The variation in error can be several orders of magnitude depending on the nature of the input data. While we are actively researching a method to inform the choice of k based on the training data, we are unable to include these results at this time. This remains a promising direction for future work, and we appreciate your suggestion

---

> > ### Comment · Reviewer_YY4J · 2024-11-27
> >
> > Thank for you the information and updates. I have updated my score.

---

> > > ### Author Response · Authors · 2024-11-27
> > >
> > > Thank you for your time in reviewing the article and your feedback!

---

### Official Review · Reviewer_vGZT · 2024-11-02

**Soundness:** 3
**Presentation:** 3
**Contribution:** 3
**Rating:** 3
**Confidence:** 2

**Summary:**

They introduce the Radial Basis Operator Network (RBON) to learn an operator in both the time and frequency domains, with adjustments to handle complex-valued inputs. Experiments on both in-distribution and out-of-distribution (OOD) data are conducted to validate RBON’s effectiveness.

**Strengths:**

They introduce the Radial Basis Operator Network (RBON) to learn an operator in both the time and frequency domains, with adjustments to handle complex-valued inputs. Experiments on both in-distribution and out-of-distribution (OOD) data are conducted to validate RBON’s effectiveness.

**Weaknesses:**

1. The experimental comparisons are insufficient, especially lacking comparisons with methods like DeepONet and FNO.
2. Using KNN to select centers may introduce instabilities.
3. The explanation of the experimental results is inadequate; for instance, Table 1 lacks clarity on why different algorithms perform variably across different problems.

**Questions:**

1. Could you elaborate on the literature review regarding learning operators in either the time domain or frequency domain?
2. What are the advantages of learning an operator simultaneously in both the time and frequency domains, as opposed to learning them separately and then combining the results?
3. Could you provide more details on the network structure?
4. How many centers are chosen using KNN, and what criteria are used for selection?
5. How do you explain the results in Table 1, where different networks show instability on different examples? LNO works better on Burgers?
6. Why some networks work better on OOD data in table 1?

---

> ### Author Response · Authors · 2024-11-26
>
> Dear Reviewer vGZT,
>
> Thank you for your constructive feedback highlighting areas for improvement in our manuscript. Below, we address your comments point by point:
>
> 1.	To the best of our knowledge FNO and its derivatives, such as [Transform Once: Efficient Operator Learning in Frequency Domain | OpenReview](https://openreview.net/forum?id=B2PpZyAAEgV), represent the primary operator networks that explicitly incorporate training in the frequency domain. FNO embeds the Fourier transform directly into its architecture, leveraging the ability to efficiently compute convolutions and capture global patterns. However, FNO also applies a linear transformation to the raw input, where parameters are learned outside of the Fourier-transformed space.
> This is different from RBON, which does not have an embedded Fourier transform. We are simply demonstrating RBONs ability to learn on data that is already in the frequency domain for the purposes of applications in the world of PDEs. Spectral methods tend to be more accurate than finite difference methods and can easily compute derivatives and convolutions in the frequency domain. Spectral methods also pick up on non-local interactions well. In contrast, operator learning in the time domain, such as in DeepONet, provides a localized perspective (like FDM), making it better suited for capturing fine-grained details and handling discontinuities or transient responses, but RBON can learn in either setting, and should thus be able to be used in either situation.
>
> 2.	This is quite an interesting point. Combining operators learned separately in the time and frequency domains could indeed enhance approximation by leveraging both global and local patterns. This is a promising direction for future research and experimentation, and we appreciate your suggestion.
> In this work, our primary motivation was to address applications in the realm of PDEs, where operators often deal with functions represented in either the frequency domain or the time domain. Our goal was to demonstrate the flexibility and capability of RBONs to effectively learn in either setting, providing a foundation for incorporating RBONs in numerical computing models.
>
> 3.	Certainly, this was a massive oversight on our part, and the revised methodology section is a much-improved description in comparison to the first version.
>
> 4.	We used the validation sets to select the optimal M and N across a very limited range (2-15) for the PDE problems to restrict the size of the network While larger networks yield better predictions, our aim was to demonstrate the accuracy achievable with smaller network configurations. Additionally, the validation sets were used to optimize hyperparameters for the other networks, ensuring a fair comparison.
>
> 5.	The performance differences can be attributed to the specific characteristics of the tasks. The Burgers equation, for example, involves smooth solutions with sharp gradients, making it well-suited to networks that excel at capturing global features (LNO, FNO, F-RBON). On the other hand, RBON and NRBON are better equipped to handle localized or transient behaviors, as seen in their performance on tasks with highly localized features or irregularities. Instability in certain networks may arise from their predisposition to either global or local patterns, which can affect their generalization, especially on OOD data. We have revised the manuscript to contain better explanations of the performance on the various PDE examples.
>
> Thank you for your insights and recommendations. They were helpful in producing a much-improved version of the paper that we hope addresses your concerns.

---

> ### Comment · Reviewer_vGZT · 2024-11-26
>
> Thanks so much for your reply! I'll keep my score. My main concerns are learning simultaneously in both the time and frequency domains seems not quite novel compared to separate cases, and the choice of k is sensitive.

---

> > ### Author Response · Authors · 2024-11-27
> >
> > Thank you for your quick response!
> >
> > To clarify further, the K-Means algorithm was used to select the centers and spreads of the RBFs, which is standard practice for RBFNs. Choosing a larger 𝑘 corresponds to a larger network, leading to improved prediction accuracy; similar to how DeepONet or other architectures benefit from being made wider or deeper. However, since RBON consists of a single layer, its complexity is determined solely by the network's width. To demonstrate RBON’s ability to accurately represent operators using small sub-networks, we intentionally limited 𝑘. For instance, predicting the wave equation required only 4 RBFs in the branch and 11 RBFs in the trunk, whereas DeepONet required multiple layers in both trunk and branch networks, each with widths exceeding 100, to achieve comparable results.
> >
> > Regarding the sensitivity, K-Means can indeed produce different results due to convergence to local extrema when initialized differently. While this introduces some variability, it can be mitigated by running K-Means multiple times with different initializations and selecting the result with the lowest within-cluster sum of squares. This straightforward approach effectively reduces any sensitivity issues due to K-Means for fixed 𝑘, and for improved accuracy 𝑘 can be increased.

---

### Official Review · Reviewer_wnkW · 2024-11-03

**Soundness:** 2
**Presentation:** 1
**Contribution:** 2
**Rating:** 3
**Confidence:** 4

**Summary:**

This paper presents an extension of DeepONet, where the DNN is replaced with a radial basis neural network, and demonstrates the performance of the extension  through solving some partial differential equations.

**Strengths:**

Replacing the DNN in DeepONet with the radial basis neural network sounds an interesting idea, leading to improved numerical results.

**Weaknesses:**

1. The presentation is unclear, making it challenging for readers unfamiliar with DeepONet to follow. For example, it lacks a clear outline (or pseudo) of the training algorithm for the proposed network. What are the training parameters? How should  M and N be selected?

2. Figure 1: what does `x' represent?  Does the linear transformation L include tunable parameters?

3. In Section 2.3, the authors mentioned computation in the frequency domain, but the algorithm is not detailed.

4. The comparison with DeepONet is not included in the paper.

5. Theoretical contribution of the paper is marginal.

**Questions:**

See weakness.

---

> ### Author Response · Authors · 2024-11-26
>
> Dear Reviewer wnkW,
>
> Thank you for your constructive feedback highlighting areas for improvement in our manuscript. Your comments have guided substantial revisions, and we believe the updated version more effectively communicates our results and methodology. Below, we address your comments point by point:
>
> 1.	Clarity and Methodology Enhancement
>
> a.	We have significantly rewritten the methodology section to make it self-contained, and accessible to readers unfamiliar with DeepONet or related structures. The training process, particularly where it diverges from standard SGD-based updates, is now explicitly detailed.
>
> b.	Regarding the selection of network widths (‘M’ and ‘N’), we acknowledge that an algorithmic methodology remains an area of ongoing research. For this paper, we employed a restricted range of possibilities to limit network size, using validation sets to guide selection within this range.
>
> 2.	Clarifications on Figure 1
>
> a.	The ‘x’ in Figure 1 is labelled in the Figure description as a Kronecker product, which is now described fully in the more detailed ‘practical implementation’ section.
>
> b.	The linear transform does not currently include tunable parameters. In fact, the implementation for this paper relies on a (tunable) parameter-free transformation for simplicity.
>
> 3.	Frequency Domain Learning
>
> The algorithm for learning in the frequency the domain is the same as learning in the time domain. The coded functions only needed to be modified to handle complex data types since we assume the data is represented in the frequency domain. The k-means algorithm required additional adaptations, particularly for inter-cluster distance calculations.
>
> 4.	Expanded Comparisons
>
> The absence of comparisons in the original submission has been thoroughly addressed. We conducted extensive numerical experiments, adding comparisons with DeepONet, FNO, LNO, and LSTM. These results highlight the strengths of RBON, particularly in terms of interpretability and accuracy, across a range of benchmarking tasks.
>
> 5.	Theoretical Contributions
>
> While the primary novelty lies in the implementation and numerical experimentation, we emphasize that RBON represents a significant step forward in scientific computing. Its small yet highly accurate architecture, which leverages interactions proportional to joint density functions, provides enhanced interpretability—a highly desirable trait for operator networks in real-world applications.
>
> We appreciate your time and effort in reviewing our paper and hope the revisions adequately address your concerns.

---

### Official Review · Reviewer_bq8u · 2024-11-04

**Soundness:** 3
**Presentation:** 3
**Contribution:** 2
**Rating:** 5
**Confidence:** 2

**Summary:**

Paper proposes a radial basis operator network, an operator network entirely represented with radial basis functions. The RBON can learn in both time and frequency domain. Experiments include comparison with Laplace neural operator on wave, burgers, and Euler-bernoulli beam equations. Paper shows RBON and NRBON forecasting global and local temperatures based on atmospheric CO2.

**Strengths:**

Paper proposes a new operator network based on radial basis functions. Paper is clearly written. The quantitative results of RBON, NRBON, and F-RBON outperform LNO on the wave, burgers, and Euler-bernoulli beam equations.

**Weaknesses:**

small set of experiments. CO2 to temperature experiment lacks baseline. other factors than CO2 affect temperature, and overall there's a lot of fluctuation in temperature, so it is hard to tell how good predictions are from the RBONs. another dataset with experiments may be helpful.

**Questions:**

perhaps authors can include the LNO results on CO2 to temperature experiment?

What is difference between RBON and NRBON? can authors clarify key architectural or mathematical differences between RBON and NRBON, and how these differences impact performance

---

> ### Author Response · Authors · 2024-11-26
>
> Dear Reviewer bq8u,
>
> Thank you for highlighting the need for a baseline in the CO2 temperature experiment, this was a clear weakness that we have addressed by running extensive computational experiments to provide better comparisons across all experiments.
>
> Extended Comparisons
>
> •	For the wave, burgers, and Euler-Bernoulli problem tasks, we have now included comparisons with FNO and DeepONet, alongside the original LNO comparisons.
> •	For the CO2 temperature experiment, we include comparisons to LSTM, LNO, FNO, and DeepONet.
>
> NRBON and RBON
>
> •	We have better described the differences between the NRBON and RBON architectures in the improved description of the network architecture and implementation.
>
> Again, we believe these revisions, based on your recommendations, significantly improve the overall description of network as well as highlighting the results.

---

### Official Review · Reviewer_xTW5 · 2024-11-04

**Soundness:** 3
**Presentation:** 1
**Contribution:** 2
**Rating:** 3
**Confidence:** 3

**Summary:**

This paper deals with operator networks working with radial basis functions.
Time and frequency domain are studied.
A couple of experiments are carried out, mainly on PDEs and TSP.

**Strengths:**

The paper is well written and the topic is interesting though not mainstream.
The experiments are relevant.

**Weaknesses:**

The methodological description of the method is too short and not detailed enough, especially for unfamiliar readers.
Some theoretical statements are made but the rest of the methodology would not allow re-implmenting the method. Figs 1 and 2 do not carry any information whatsoever; most equations about the network are implicitly embedded in the theorem.
Part of the problem is that the paper describes a variant of NO/ON and the authors focus on the increment, probably for the sake of space, at the expense of making the paper difficult to read for the average reader.
With the amount of provided details, the proposed method seems to be very simple and very close to classical RBFNs with a product on top. The PDE and TSP tasks are not very convincing if they are not compared to other architecture and methods (X-RBON get compared to LNO only).

**Questions:**

Can you please reinforce the methodological part and make more self-contained and more detailed.
Can you extend the comparisons (if you think it makes sense)?

---

> ### Author Response · Authors · 2024-11-26
>
> Dear Reviewer xTW5,
>
> Thank you for highlighting the weaknesses regarding the methodology section. We appreciate your comments emphasizing the need for greater clarity and comprehensiveness. In response, we have made the following changes:
>
> Expanded the Methodology Section
>
> •	We have significantly rewritten the methodology section to make it self-contained, providing detailed, step-by-step instructions in the practical implementation section.
> •	The architecture, particularly how it extends beyond classical RBFNs, is more clearly described.
>
> Improved Figure Description
>
> •	The description accompanying Figure 1 better details the type of product between the two subnetworks.
>
> Extended Comparisons
>
> •	We appreciate the advice to extend comparisons and have completed extensive experimentation in response. For the PDE tasks, we have now included comparisons with FNO and DeepONet, alongside the original LNO comparisons.
> •	 For the TSP task, we have expanded the benchmarks to include LSTM, LNO, FNO, and DeepONet.
>
> We believe these revisions, based on your recommendations, significantly improve the overall description of network as well as highlighting the results.

---

### Note · Authors · 2025-02-27

I have read and agree with the venue's withdrawal policy on behalf of myself and my co-authors.

---

### Meta-Review · Area_Chair_KJ7Y · 2024-12-21

**Metareview:**

The paper proposes a novel (radial basis) operator network capable of  learning nonlinear operators in both time and frequency domains, particularly when
handling complex-valued inputs.
Several critical points (presentation, experiments, baselines,...) have been raised by the reviewers and some of this issue have been
addressed by the authors during the rebuttals and in the revision.  However, there is still some concerns about novelties
and experimental analysis of the model making it hard to accept the paper in
its current status.

**Additional Comments On Reviewer Discussion:**

Reviewers acknowledged the rebuttals but concerns have not been cleared

---

### Decision · Program_Chairs · 2025-01-22

Reject